# Rebamipide ameliorates indomethacin-induced small intestinal damage and proton pump inhibitor-induced exacerbation of this damage by modulation of small intestinal microbiota

**Tetsuya Tanigawa**[1,2]*, **Toshio Watanabe**[1], **Akira Higashimori**[1], **Sunao Shimada**[1,2], **Hiroyuki Kitamura**[1], **Takuya Kuzumoto**[1], **Yuji Nadatani**[1], **Koji Otani**[1], **Shusei Fukunaga**[1], **Shuhei Hosomi**[1], **Fumio Tanaka**[1], **Noriko Kamata**[1], **Yasuaki Nagami**[1], **Koichi Taira**[1], **Masatsugu Shiba**[1], **Wataru Suda**[3], **Masahira Hattori**[3], **Yasuhiro Fujiwara**[1]

1 Department of Gastroenterology, Osaka City University Graduate School of Medicine, Osaka, Japan,
2 Department of Gastroenterology, Osaka City Juso Hospital, Osaka, Japan, 3 Laboratory for Microbiome Sciences, Center for Integrative Medical Sciences, RIKEN, Kanagawa, Japan

* ttanigawa@med.osaka-cu.ac.jp

**Data Availability Statement:** All relevant data are available in figshare

## Abstract

Non-steroidal anti-inflammatory drugs (NSAIDs) induce small intestinal damage. It has been reported that rebamipide, a mucoprotective drug, exerts a protective effect against NSAID-induced small intestinal damage; however, the underlying mechanism remains unknown. In this study, we investigated the significance of the small intestinal microbiota in the protective effect of rebamipide against indomethacin-induced small intestinal damage in mice. A comprehensive analysis of the 16S rRNA gene sequencing revealed an alteration in the composition of the small intestinal microbiota at the species level, modulated by the administration of rebamipide and omeprazole. The transplantation of the small intestinal microbiota of the mice treated with rebamipide suppressed the indomethacin-induced small intestinal damage. Omeprazole, a proton pump inhibitor, exacerbated the indomethacin-induced small intestinal damage, which was accompanied by the alteration of the small intestinal microbiota. We found that the transplantation of the small intestinal microbiota of the rebamipide-treated mice ameliorated indomethacin-induced small intestinal damage and the omeprazole-induced exacerbation of the damage. These results suggest that rebamipide exerts a protective effect against NSAID-induced small intestinal damage via the modulation of the small intestinal microbiota, and that its ameliorating effect extends also to the exacerbation of NSAID-induced small intestinal damage by proton pump inhibitors.

## Introduction

The emergence of video capsule endoscopy and balloon-assisted enteroscopy have enabled the precise diagnosis of diseases involved in small intestinal damage, and the clinical features of

(https://figshare.com/articles/dataset/
Rebamipide_Suppress_Indomethacin-Induced_
Small_Intestinal_Damage_by_Modulation_of_
Small_Intestinal_Microbiota/13241267), DOI: 10.
6084/m9.figshare.13241267.

**Funding:** The authors received no specific funding
for this work.

**Competing interests:** The authors have declared
that no competing interests exist.

nonsteroidal anti-inflammatory drug (NSAID)-induced small intestinal damage have been
clarified [1–6]. NSAIDs frequently induce small intestinal damage, which cause overt and
occult bleeding, perforation, and stenosis of the small intestine [2, 7]. Our previous clinical
study revealed that 25% of the chronic NSAID users from among the patients with rheumatoid
arthritis had mild damage; more importantly, 27.8% had severe damage and significantly
decreased hemoglobin levels [8].

Accumulating clinical evidence suggests that proton pump inhibitors (PPIs) exacerbate
NSAID-induced small intestinal damage. PPIs are frequently prescribed with NSAIDs for pro-
phylactic therapy against NSAID-induced upper gastroduodenal ulcers and bleeding [9]. How-
ever, our previous study has suggested that the use of PPIs was an independent risk factor for
severe NSAID-induced small intestinal damage in patients with rheumatoid arthritis who
were chronic NSAID users [8]. Endo and his colleagues also showed that PPIs were risk factors
for low-dose aspirin-induced small intestinal damage [10]. A recent randomized, placebo-con-
trolled trial in healthy volunteers revealed that rabeprazole increased the incidence and num-
ber of small intestinal damage induced by celecoxib [11]. Basic research has also demonstrated
the deleterious effect of PPIs on NSAID-induced small intestinal damage [12–14].

The induction of dysbiosis in the small intestine by PPI is proposed as a possible mecha-
nism by which PPI exacerbates NSAID-induced small intestinal damage. Dysbiosis is an
abnormality in the composition of the microbial community in which the population of bacte-
ria beneficial for the host health decreases and the population of pathogenic bacteria, typically
present in small numbers, increases [15]. Wallace and his colleagues demonstrated, in basic
experimental research, that PPI exacerbate NSAID-induced small intestinal damage in rats,
and that this is accompanied by the induction of dysbiosis by PPIs [13, 16]. Conversely, the
beneficial modulation of the small intestinal microbiota so that it is resistant to NSAID-
induced small intestinal damage could be a good strategic option for the prevention and ther-
apy of NSAID-induced small intestinal damage.

Rebamipide [(2 RS)-2-(4-chlorobenzoylamino)-3-(2oxo-1,2-dihydroquinolin-4-yl) propa-
noic acid] is a mucoprotective drug that has been used clinically for the treatment of gastritis
and peptic ulcers [17]. Recent clinical evidence indicates that rebamipide exerts an ameliorat-
ing effect on NSAID-induced small intestinal damage [18–21]. Previous studies, including one
by our research group, revealed that rebamipide suppressed indomethacin-induced small
intestinal damage, which was accompanied by the alteration of the intestinal microbiota [22–
24]. However, in the previous studies, it was not determined if the rebamipide-modulated
small intestinal microbiota has the potential to ameliorate NSAID-induced small intestinal
damage. It also remained unclear whether rebamipide suppressed the exacerbation of NSAID-
induced small intestinal damage by PPIs.

In the present study, we aimed to establish the following in NSAID-induced small intestinal
damage: i) whether rebamipide suppress the NSAID-induced small intestinal damage directly
via the modulation of the small intestinal microbiota; and ii) the effect of rebamipide on the
exacerbation of NSAID-induced small intestinal damage by PPIs via the modulation of the
small intestinal microbiota. We found that rebamipide ameliorates indomethacin-induced
small intestinal damage and omeprazole-induced exacerbation of this damage via the modula-
tion of the small intestinal microbiota.

## Materials and methods

### Induction of small intestinal damage by indomethacin in mice

Seven-week-old specific-pathogen-free male C57BL/6 mice were purchased from Charles
River Japan, Inc. (Yokohama, Japan). All animals were housed in polycarbonate cages with

paper chip bedding in the filtered-air ventilated cage rack of an air-conditioned biohazard room with a 12-h light-dark cycle. All animals had free access to food and water. The animals were fed with standard rodent diet (CE-2; CLEA Japan Inc.,Tokyo, Japan). The animals of treatment groups were separately kept in individual polycarbonate cages after shipping through the study.

When performing invasive procedures including intravenous injection and euthanasia, anesthesia was always conducted using isoflurane anesthetizer (MK-A110D, Muromachi Kikai, Tokyo, Japan). Anesthesia was induced at 4% isoflurane with 20% oxygen using a poly-carbonate chamber and maintained at 1.5–2.75% isoflurane with 20% oxygen using anesthetic mask. Euthanasia was performed by instant cervical dislocation under deep sedation. This study was carried out in strict accordance with the recommendations in the Guide for the Care and Use of Laboratory Animals of the National Institutes of Health. All experiments were carried out with confirmation of The Regulations on Animal Experiments and with approval of The Institutional Animal Care and Use Committee of Osaka City University Graduate School of Medicine (protocol ID: 09037). All invasive procedures were performed under isoflurane anesthesia, and all efforts were made to minimize suffering.

To induce small intestinal damage, 10 mg/kg of indomethacin (Sigma Chemical Company, St. Louis, MO) with vehicle (0.5% carboxymethylcellulose) was orally administered by gavage with intragastric feeding tube to non-fasted animals. For the evaluation of macroscopic damage, 1% Evans blue was injected intravenously 30 min before sacrifice in order to delineate the mucosal damage; after sacrifice, the small intestine was collected and opened along the antimesenteric attachment side of the lumen. The macroscopic mucosal damage was defined as ulcer or erosion clearly delineated by Evans blue dye. The macroscopic mucosal damage was measured in a masked fashion by the investigator (A. H.). The shape of macroscopic mucosal damage was round or elliptical shape, therefore we measured the major axis and minor axis with a digital precision caliper, multiplied major and minor axis, summed for each of the small intestine, and used the value as the lesion index. For histological evaluation, each of the small intestinal tissue samples that exhibited typical mucosal damage was fixed with 10% buffered formalin, and 4-μm thick tissue sections were mounted on glass slides, and subjected to hematoxylin and eosin (H&E) staining.

## Histological evaluation of small intestinal damage

Tissue sections stained with H&E were viewed under high power using a white-light microscope. For each mouse, at least 10 random villi in the injured areas were scored independently in a masked fashion by two investigators (H.K. and T.K.). For evaluation, a modified histological scoring system was used [25]. The histological score ranged from 0 to 13 and was divided into the following six categories: epithelium (0 = normal, 1 = flattened, 2 = loss of epithelial continuity, 3 = severe denudation), villus shape (0 = normal, 1 = short and rounded, 2 = extremely short and thick), villus tip (0 = normal, 1 = damaged, 2 = severely damaged), stroma (0 = normal, 1 = slightly retracted, 2 = severely retracted), inflammation (0 = no infiltration, 1 = mild infiltration, 2 = severe infiltration), and crypt status (0 = normal, 1 = mild crypt loss, 2 = severe crypt loss).

## Experimental groups and transplantation of small intestinal microbiota

Donor mice for the transplantation of the small intestinal microbiota received oral administration of vehicle (0.5% carboxymethylcellulose) or rebamipide (300 mg/kg body weight), which was supplied by Otsuka Pharmaceutical Co. (Tokyo, Japan) once daily for 1 week.

Twenty-four hours after the final drug administration, the animals were sacrificed, and the contents of the ileum were collected.

In the first series of the experiments, the recipient mice were treated with antibiotics (penicillin/streptomycin/ampicillin) for 7 d. Penicillin and streptomycin (Sigma-Aldrich Co. LCC., Japan) were given in bottle water (penicillin: 6000 U/L of water, streptomycin 0.6 g/L of water) and ampicillin [800 mg/kg body weight (BW), Sigma-Aldrich] was administered once per day by gavage with intragastric feeding tube. Twenty-four hours after the last administration of ampicillin, the water bottle containing penicillin and streptomycin was switched for a bottle of pure water, and the mice were subjected to the transplantation of small intestinal microbiota by the oral administration of the ileal content obtained from vehicle- or rebamipide-treated mice. The content of the ileal lumen was suspended in normal saline at a concentration of 100 mg/mL; the suspension was centrifuged at 1,000 rpm for 5 min and the supernatant was collected. The mice were administered 0.4 mL of the supernatant by gavage with intragastric feeding tube. After 5 d, the mice were administered 10 mg/kg of indomethacin and the intestinal damage was evaluated 24 h after administration of indomethacin.

In the second series of the experiments, to assess the influence of omeprazole on indomethacin-induced small intestinal damage and the effect of rebamipide-modulated microbiota on the exacerbation of indomethacin-induced small intestinal damage by omeprazole, the mice were treated with the same antibiotics as the first series of experiments for 7 d. At 24 h after the final antibiotic treatment, the mice were divided into three groups: (1) the mice transplanted with small intestinal microbiota obtained from vehicle-treated mice and administered with vehicle for 5 d; (2) the mice transplanted with small intestinal microbiota obtained from vehicle-treated mice and administered with omeprazole (140 mg/kg BW, FUJIFILM Wako Pure Chemical Corporation, Osaka, Japan) for 5 d; and (3) the mice transplanted with small intestinal microbiota obtained from rebamipide-treated mice and administered with omeprazole (140 mg/kg BW) for 5 d. After 5 d, the mice in all groups were administered indomethacin (10 mg/kg BW) and 24 h later the intestinal damage was evaluated. Omeprazole and indomethacin were administered by gavage with an intragastric feeding tube.

The protocol is summarized in Fig 1.

## Sample collection and DNA extraction

The content of the small intestinal lumen was collected immediately after removing the intestine from the mice. The samples were frozen with liquid nitrogen and stored at −80˚C until use. In brief, prior to DNA extraction, each sample was suspended in 15 mL PBS buffer (Thermo Fisher Scientific K.K., Tokyo, Japan) and sample suspension was filtered through a 100-μm mesh nylon filter (Corning Inc., New York, NY, USA) to remove eukaryotic cells and other debris. The debris on the filter was washed twice using a plastic bar with PBS buffer. The bacteria-enriched pellet was obtained by centrifugation of the filtrate at 9,000 $g$ for 10 min at 4˚C. The pellet was washed once with 35 mL PBS, and once again with TE20 buffer (10 mM Tris-HCl (Sigma-Aldrich Co. LCC., Tokyo, Japan), 20 mM EDTA (Thermo Fisher Scientific K.K., Tokyo, Japan)); then, DNA was extracted.

High molecular weight DNA was extracted from the samples by enzymatic lysis [26, 27]. The bacterial pellets suspended in 35 mL TE20 were centrifuged at 9,000 $g$ for 10 min at 4˚C. The pellets were resuspended in 0.4 mL TE20 and the suspension was incubated with a final concentration of 15 mg/mL lysozyme (Sigma-Aldrich Co. LCC.) and 2,000 units/mL of purified achromopeptidase (FUJIFILM Wako Pure Chemical Co., Osaka, Japan) for 2 h at 37˚C with gentle shaking. The mixture was further incubated in the solution to a final concentration of 1% SDS (Sigma-Aldrich Co. LCC.) and 1 mg/mL of proteinase K (Merck & Co. Inc., USA)

# Experiment 1

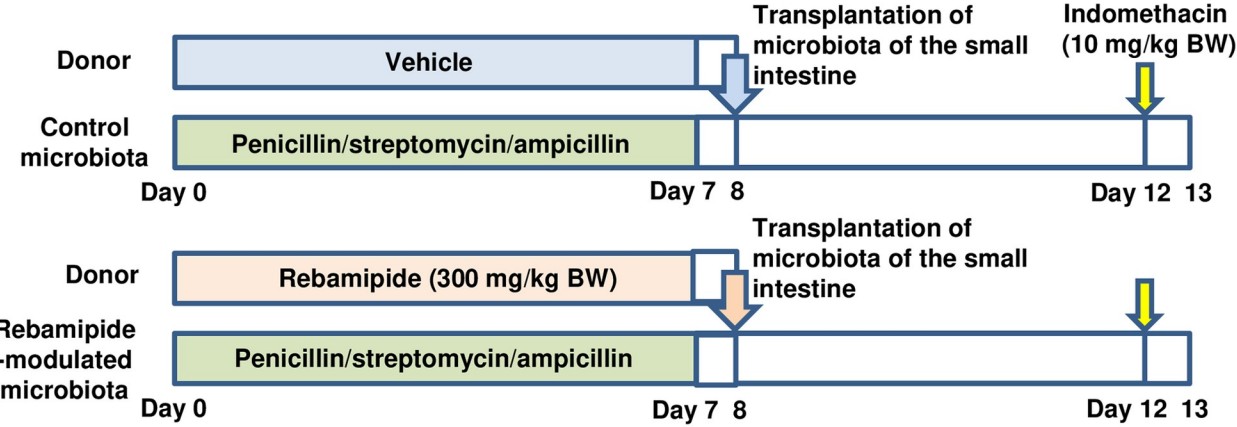

# Experiment 2

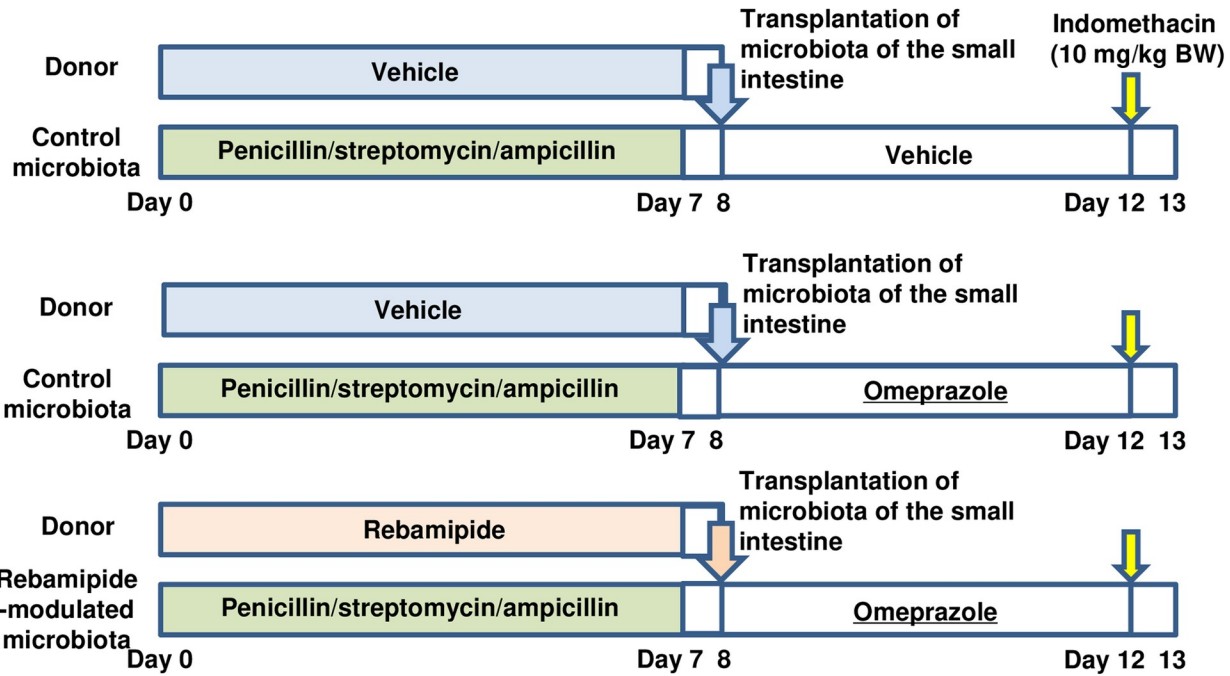

**Fig 1. Schemas of design of experimental protocol.**

for 1 h at 55˚C. The lysate was gently mixed with an equal volume (500 μL) of phenol/chloroform/isoamyl alcohol (Nippon Gene Co. Ltd., Tokyo, Japan) for 10 min, and the mixture was centrifuged at 9000 $g$ (12000 rpm) for 10 min at room temperature. The aqueous supernatant was transferred to a new clean tube, and DNA was precipitated by the addition of 3 M sodium

acetate (Nippon Gene Co. Ltd.) to the final concentration of 0.3 M, followed by the addition of an equal volume of isopropanol (FUJIFILM Wako Pure Chemical Co.). The DNA was pelleted by centrifugation at 9000 $g$ (12000 rpm) for 10 min at 4˚C, rinsed once with 75% ethanol, and dissolved in 500 μL of TE buffer. After the DNA was treated with 10 mg/mL RNase A (Nippon Gene Co. Ltd.) for 30 min at 37˚C, 0.6 volumes of PEG solution (20% polyethylene glycol 6000 and 2.5 M NaCl; Hampton Research Corp., Aliso Viejo, CA, U.S.) was added, mixed gently, and kept on ice for at least 10 min. The DNA was pelleted by centrifugation at 9000 $g$ (12000 rpm) for 10 min at 4˚C, rinsed twice with 75% ethanol twice, and dissolved in 50 μL of TE buffer. The DNA concentration was measured using a Qubit Broad Range and High Sensitivity (Thermo Fisher Scientific K.K.).

## 16S rRNA gene sequencing

The 16S rRNA analysis of DNA samples was performed as previously described [27]. Briefly, PCR was performed using 27Fmod 5′–AGRGTTTGATYMTGGCTCAG–3′ and 338R 5′– TGCTGCCTCCCGTAGGAGT–3′ to amplify the V1–V2 region of the 16S rRNA gene [26]. The amplified DNA samples (~330 bp) were subsequently purified using AMPure XP (Beckman Coulter, Brea, CA, USA), and quantified using a Quant-iT Picogreen dsDNA assay kit (Invitrogen, Waltham, MA, USA) and a TBS-380 Mini-Fluorometer (Turner Biosystems, Waltham, MA, USA). The 16S sequencing was performed using MiSeq in accordance with the Illumina protocol (Illumina, San Diego, CA). The paired-end reads were merged using the fastq-join program based on overlapping sequences. Reads with an average quality value of <25 and inexact matches to both universal primers were filtered out.

Filter-passed reads were used for further analysis after trimming off both primer sequences. For each sample, the quality filter-passed reads were rearranged in descending order in accordance with the quality value and then clustered into OTUs with a 97% pairwise-identity cutoff using the UCLUST program version 5.2.32 (https://www.drive5.com). Taxonomic assignment of each OTU was made by a similarity search of the Ribosomal Database Project (RDP) and the National Center for Biotechnology Information (NCBI) genome database using the GLSEARCH program. For calculation of microbial abundance, taxa with a relative abundance of >0.1% were considered as positive. For assignment at the phylum, genus, and species levels, the sequence similarity thresholds of 70%, 94%, and 97%, were applied, respectively [26, 28, 29].

The UniFrac distance was used for the assessment of the dissimilarity (distance) between any pair of the samples [30]. We performed a principal coordinate analysis (PCoA) to visualize the similarities or dissimilarities in the microbiome structure in the UniFrac analysis using skbio.stats.ordination module of Python. We conducted Permutational multivariate analysis of variance (PERMANOVA, adonis in the R-vegan package) to compare the overall microbiome structure, and the $p$-values were adjusted for multiple testing by the Benjamin–Hochberg procedure. We used the observed and Chao 1-estimated OTU numbers and Shannon's index to evaluate the species richness and diversity of the overall microbial community. Chao1 and Shannon's index were calculated using Scikit-Bio's diversity function.

The similarity of the relative abundance at the phylum, family, genus, and species levels was assessed using the Kruskal–Wallis test followed by the Steel–Dwass test for multiple comparisons.

## Statistical analysis

The data are presented as the mean ± standard error of the mean (S.E.M.). The Mann–Whitney $U$ test was used to evaluate the between-group differences. The differences among multiple

groups were first analyzed by the Kruskal–Wallis test, and when a statistical significance was detected, The Mann–Whitney *U* test with Bonferroni correction was used to determine the statistical significance between multiple testing groups. Statistical significance was set at *p<0.05.*

## Results

### The effects of transplantation of rebamipide-modulated small intestinal microbiota on indomethacin-induced small intestinal damage

We determined whether the small intestinal microbiota derived from the mice treated with rebamipide exert inhibitory effects on the indomethacin-induced small intestinal damage. As shown in Fig 2, the lesion index of the mice transplanted with the small intestinal microbiota obtained from rebamipide-treated mice was smaller than that of the mice transplanted with the small intestinal microbiota obtained from vehicle-treated mice (Fig 2B). The histological scores of the vehicle and rebamipide groups treated with indomethacin showed a similar tendency with respect to the lesion indices (Fig 2C).

### The influence of omeprazole on small intestinal damage and the inhibitory effect of transplantation of the rebamipide-modulated small intestinal microbiota on exacerbation of indomethacin-induced small intestinal damage by omeprazole

Next, we examined whether omeprazole exacerbate indomethacin-induced small intestinal damage in mice and whether it is suppressed by modulation of the small intestinal microbiota by rebamipide. In mice transplanted with the small intestinal microbiota obtained from vehicle-treated mice, the administration of omeprazole increased the lesion index (Fig 3B). In mice administered with omeprazole, the lesion index in mice transplanted with the small intestinal microbiota obtained from rebamipide-treated mice was smaller than that in mice transplanted with the small intestinal microbiota obtained from vehicle-treated mice (Fig 3B). The histological scores of these groups treated with indomethacin showed a similar tendency with respect to the lesion indices (Fig 3C).

### The effect of rebamipide on the composition of the small intestinal microbiota in mice

There was no difference in OTU number in the small intestinal microbiota between the control- and rebamipide-treated groups. Shannon's index, ACE and the Chao1 index were also similar between the control- and rebamipide-treated groups (Table 1).

UniFrac distances between the small intestinal microbial communities of the mice were visualized by a scatter plot created by PCoA to assess the β-diversity (Fig 4). PERMANOVA of the unweighted UniFrac distances showed significant differences in the composition of the small intestinal microbiota between the control- and rebamipide-treated groups. PERMANOVA of the weighted UniFrac distances showed the trend in the differences in the composition of the small intestinal microbiota between the control- and rebamipide-treated groups (Table 2).

At the phylum level, the majority of the small intestinal microbiota in the mice was dominated by *Firmicutes* followed by *Bacteroidetes* and *Proteobacteria*. There was no significant difference in taxonomic assignment performed at the phylum level between the control- and rebamipide-treated groups (Fig 5A).

**A**

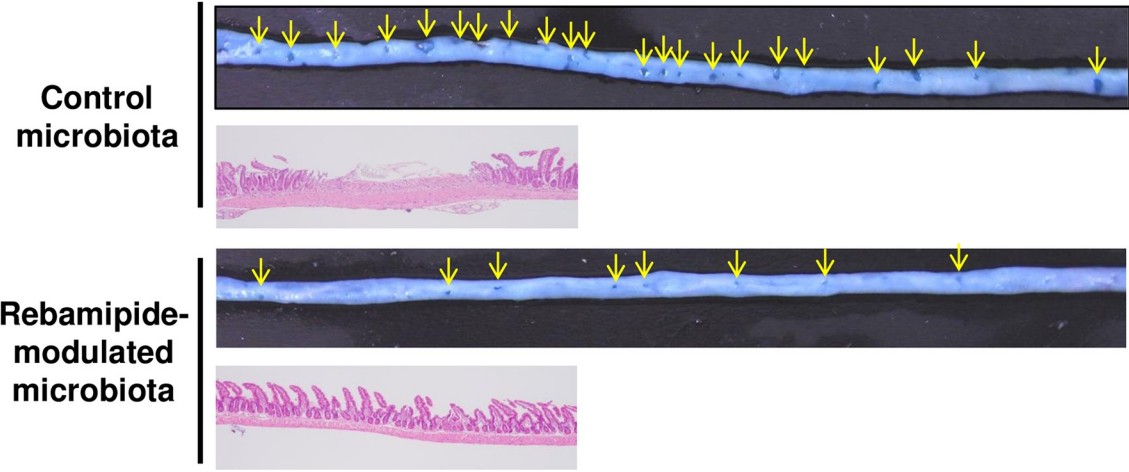

Control microbiota

Rebamipide-modulated microbiota

**B**

**C**

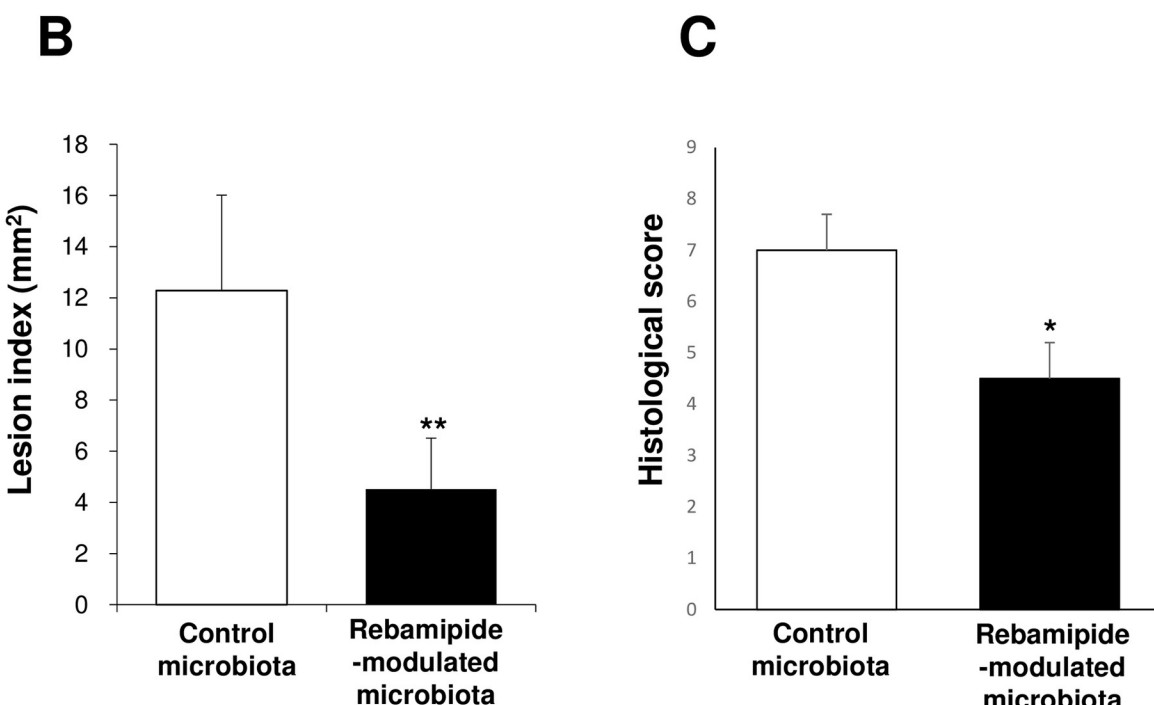

**Fig 2. Small intestinal damage after indomethacin administration and the inhibitory effects of rebamipide-modulated small intestinal microbiota on indomethacin-induced small intestinal damage.** A. Typical macroscopic and histological image of indomethacin-induced small intestinal damage. Damaged mucosa (arrows) was stained dark blue with 1% Evans blue. B. Lesion indices of indomethacin-induced small intestinal damage. C. Histological evaluation of indomethacin-induced small intestinal damage. Ct: control microbiota group Reb: rebamipide-modulated microbiota group. Each column represents the mean ± standard error of the mean (S.E.M.). $N$ = 5–6. $^*p$<0.05 and $^{**}p$< 0.01 vs. control microbiota group.

**A**

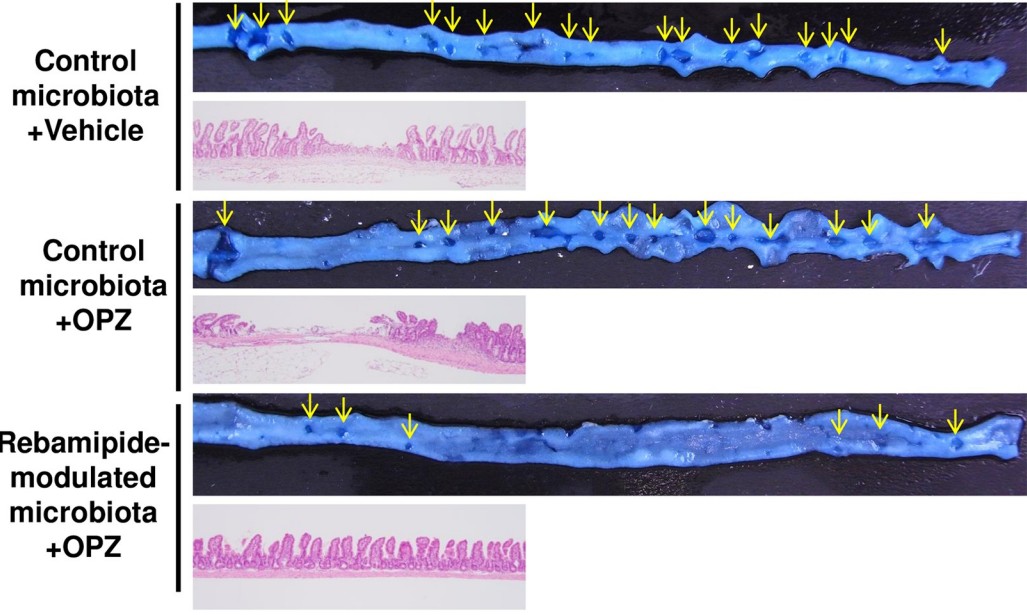

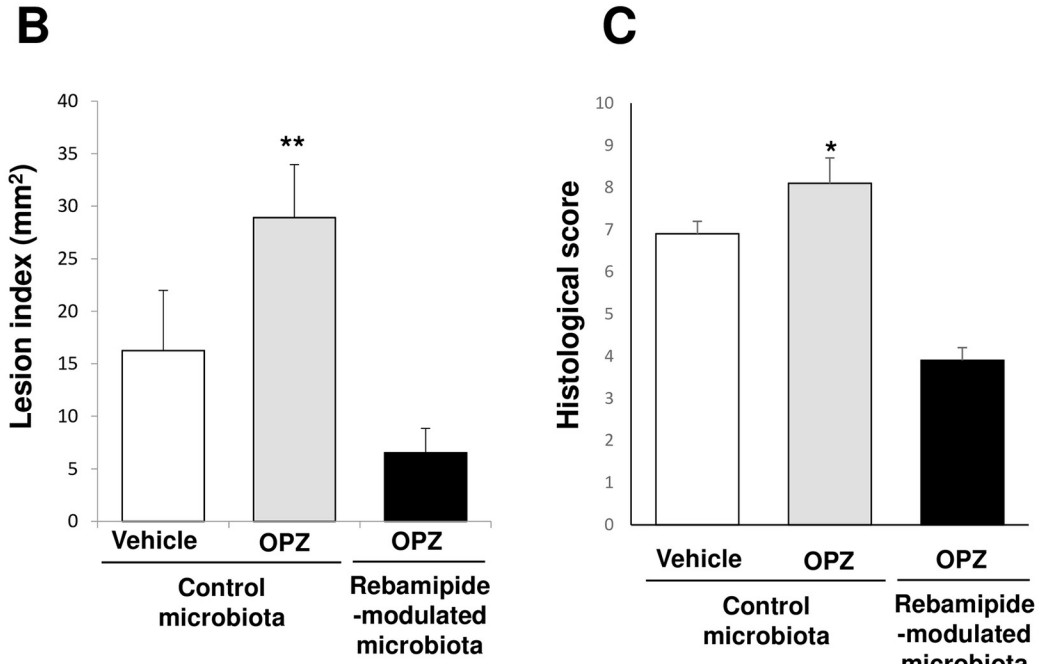

**Fig 3. The deleterious effect of omeprazole and the inhibitory effects of rebamipide-modulated small intestinal microbiota on indomethacin-induced small intestinal damage.** A. Typical macroscopic and histological image of indomethacin-induced small intestinal damage. Damaged mucosa (arrows) was stained dark blue with 1% Evans blue. B. Lesion indices of indomethacin-induced small intestinal damage. C. Histological score of indomethacin-induced small intestinal damage. OPZ: omeprazole Each column represents the mean ± standard error of the mean (S.E.M.). $N$ = 6–7. [*]$p$<0.05 and [**]$p$< 0.01 vs. vehicle-administered mice transplanted with control microbiota.

**Table 1. α-diversity, measured by operational taxonomic unit (OTU) number, Chao1 index, ACE index, and Shannon's diversity index of small intestinal lumen contents in mice given rebamipide or vehicle.**

| Index | control microbiota | rebamipide-modulated microbiota |
|---|---|---|
| Number of OTU | 107.62 ± 15.94 | 108.88 ±14.33 |
| Chao1 | 242.70 ± 36.55 | 252.25 ± 41.51 |
| ACE | 238.74 ± 39.63 | 265.61 ± 43.72 |
| Shannon | 2.47 ± 0.31 | 2.52 ± 0.23 |

$N$ = 8. Values are expressed as mean ± standard error of the mean (S.E.M.).

Taxonomic assignment performed at the genus level showed that the major constituent of the small intestinal microbiota was the genus *Lactobacillus*. There was a statistically significant difference in *Dubosiella* and *Enterococcus* between the control- and rebamipide-treated groups; however, these bacteria constituted a minor proportion of the population of the small intestinal microbiota (Fig 5B).

At the species level, the dominant species in small intestinal microbiota was *L. taiwanensis* followed by *L. reuteri* and *L. murinus*. The administration of rebamipide decreased the proportion of *L. taiwanensis*, but increased the proportion of *L. murinus* (Fig 5C).

*Bacteroidetes* to *Firmicutes* ratio (B/F ratio) is a parameter of dysbiosis in the large intestine and it is indicated that the ratio is associated with some pathophysiological conditions such as obesity [31], diabetes [32], and multiple sclerosis [33]. In the present study there was not significant difference in B/F ratio in the small intestinal microbiota between the control- and rebamipide-treated groups ($p$ = 0.17) (Fig 5D).

## Influence of omeprazole on the composition of small intestinal microbiota in mice transplanted with the control small intestinal microbiota and mice transplanted with rebamipide-modulated small intestinal microbiota

In mice transplanted with small intestinal microbiota from vehicle-treated animals, the administration of omeprazole significantly decreased Chao1 index, ACE index, and Shannon's index of the small intestinal microbiome. Transplantation with small intestinal microbiota from rebamipide-treated mice reversed these parameters observed in omeprazole-treated mice transplanted with small intestinal microbiota from vehicle-treated mice, up to the values observed in the vehicle-treated mice (Table 3).

PERMANOVA of both unweighted and weight UniFrac distances showed significant differences between the vehicle-administered group transplanted with small intestinal microbiota obtained from control mice, the omeprazole-administered group transplanted with small intestinal microbiota obtained from vehicle-treated mice, and the omeprazole-administered group transplanted with small intestinal microbiota obtained from rebamipide-treated mice (Table 2).

At the phylum level, the administration of omeprazole decreased the population of *Bacteroidetes*, whereas the influence of omeprazole on *Bacteroidetes* was ameliorated in mice transplanted with the small intestinal microbiota from rebamipide-treated mice. The administration of omeprazole decreased the population of *Actinobacteria*, however, *Actinobacteria* was a minor population (less than 1%) (Fig 6A).

At the genus level, the administration of omeprazole tended to increase the population of *Lactobacillus* and significantly decreased the population of *Robinsoniella* (Fig 6B).

At the species level, the administration of omeprazole increased the population of *L. taiwanensis*, whereas the influence of omeprazole on the population of *L. taiwanensis* was

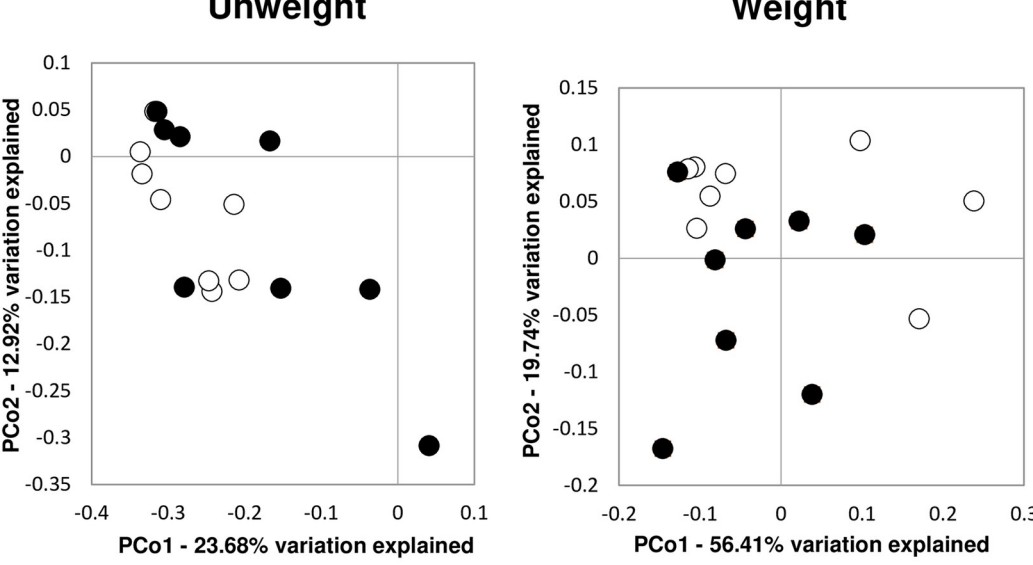

○: Control microbiota
●: Rebamipide-modulated microbiota

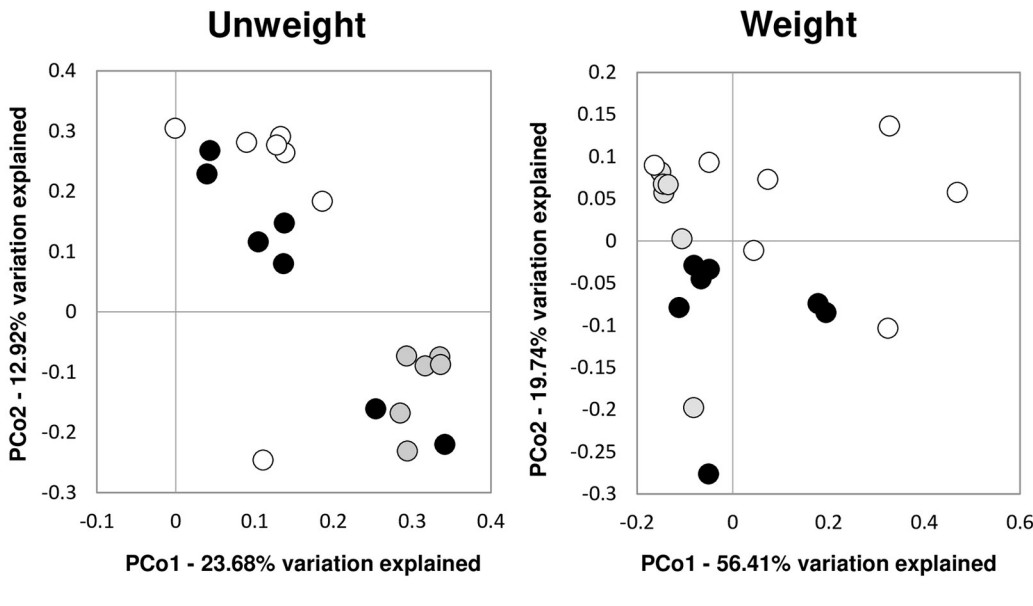

○: Control microbiota + vehicle
◐: Control microbiota + omeprazole
●: Rebamipide-modulated microbiota + omeprazole

**Fig 4. Principal coordinate analysis (PCoA) of unweighted and weighted UniFrac distances of small intestinal microbiota.**

**Table 2. PERMANOVA of UniFrac distance in small intestine.**

| Category | Weigthed UniFrac | | Unweigthed UniFrac | |
|---|---|---|---|---|
| | $R^2$ | *P* value | $R^2$ | *P* value |
| control microbiota vs rebamipide-modulated microbiota | 0.14909 | 0.06494 | 0.12584 | 0.000999 |
| control microbiota+vehicle vs control microbiota+omeprazole | 0.3884 | 0.01698 | 0.28218 | 0.002997 |
| control microbiota+omeprazole vs rebamipide-modulated microbiota + omeprazole | 0.33983 | 0.005994 | 0.20049 | 0.00999 |
| control microbiota+vehicle vs rebamipide-modulated microbiota + omeprazole | 0.2165 | 0.06993 | 0.1494 | 0.01499 |

ameliorated in mice transplanted with the small intestinal microbiota from rebamipide-treated mice. The population of *L. murinus* was not significantly affected by administration of omeprazole, but was significantly increased by rebamipide in mice transplanted with the small intestinal microbiota from rebamipide-treated mice (Fig 6C).

Administration of omeprazole decreased the B/F ratio, while the influence of omeprazole on B/F ratio was ameliorated in mice transplanted with the small intestinal microbiota from rebamipide-treated mice (Fig 6D).

## Discussion

In the present study, we demonstrated that rebamipide ameliorated indomethacin-induced small intestinal damage, as previously reported [22]. Moreover, by using transplantation with the small intestinal microbiota, we demonstrated that the effect of rebamipide on indomethacin-induced small intestinal damage was, at least in part, due to the modulation of the small intestinal microbiota. Finally, we revealed that small intestinal microbiota from mice treated with rebamipide exerted inhibitory effect against the exacerbation of indomethacin-induced small intestinal damage by omeprazole. These results suggest that the small intestinal microbiota modulated by rebamipide results in resistance to exacerbation of NSAID-induced small intestinal damage by PPI.

Recently, several clinical studies, including our previous study, suggest that PPI is a risk factor for NSAID-induced small intestinal damage [8, 10, 11]. It is speculated that the possible mechanism through which PPI exacerbate NSAID-induced small intestinal damage is the induction of dysbiosis in the small intestine. Several studies suggest that the loss of microbial diversity in the gut microbiota is associated with some gastrointestinal diseases, such as Crohn's disease [34], ulcerative colitis [35], irritable bowel syndrome [36], graft-versus-host disease in allogeneic hematopoietic-cell transplantation [37–40]. Moreover, it is reported that lower microbial diversity correlated with increased small intestinal permeability in chronic liver diseases [41]. Interestingly, we recently reported that PPI significantly enhanced the stress-induced pathogenic increase in small intestinal permeability accompanied by reduction of α-diversity [42]. The present study showed that omeprazole tended to decrease the parameters of microbial diversity, OTU number, Chao1 index, and Shannon index. The result was similar to a previous clinical study, in that PPI use was associated with a significant decrease in Shannon's diversity in the gut microbiome [43]. These results suggest that PPI induce the loss of small intestinal microbial diversity, which may be a crucial factor for the exacerbation of indomethacin-induced small intestinal damage by PPI. Importantly, mice transplanted with small intestinal microbiota obtained from mice treated with rebamipide showed resistance to the omeprazole-induced loss of microbial diversity, which may account for the mechanism through which rebamipide ameliorate the exacerbation of NSAID-induced small intestinal damage by PPI via modulation of the small intestinal microbiota.

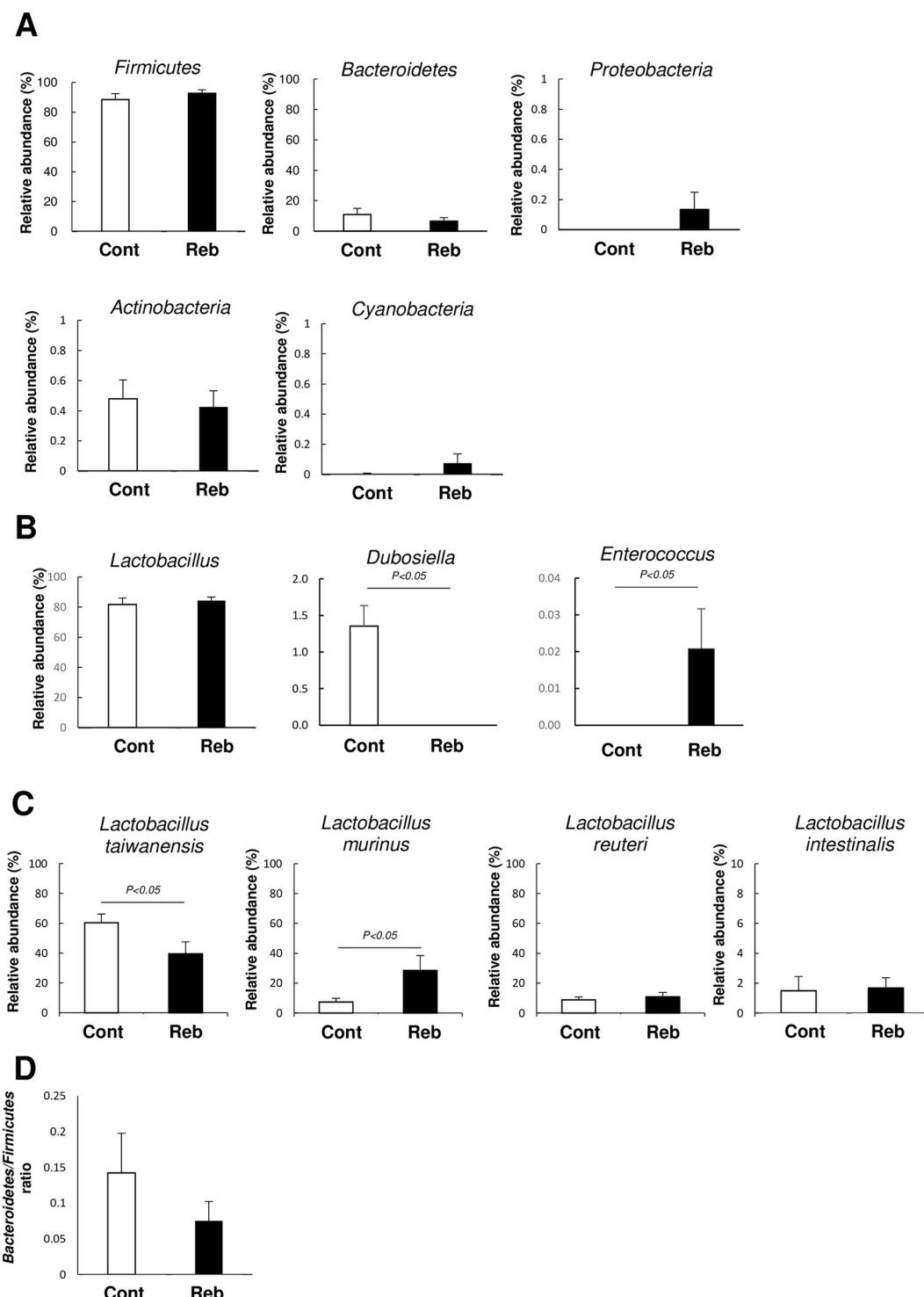

**Fig 5.** The major bacterial composition of small intestine in mice given rebamipide or vehicle at (A) phylum level (B) genus level (C) species level. (D) *Bacteroidetes* to *Firmicutes* ratio. Cont: control group Reb: rebamipide-treated group. Each column represents the mean ± standard error of the mean (S.E.M.). *N* = 6–7.

**Table 3. α-diversity, measured by operational taxonomic unit (OTU) number, Chao1 index, ACE index, and Shannon's diversity index of small intestinal lumen contents in microbiota-transplanted mice administered with omeprazole.**

| Index | control microbiota + vehicle | control microbiota + omeprazole | rebamipide-modulated microbiota + omeprazole |
|---|---|---|---|
| Number of OTUs | 92.86 ± 13.68 | 51.33 ± 7.77 | 91.00 ± 16.69 |
| Chao1 | 208.63 ± 37.48 | 97.68 ± 22.10[*] | 190.68 ± 31.10[#] |
| ACE | 199.66 ± 27.74 | 98.60 ± 20.40[*] | 180.41 ± 38.84 |
| Shannon | 2.69 ± 0.20 | 1.53 ± 0.21[**] | 2.65 ± 0.23[##] |

$N$ = 6–7. Values are expressed as mean ± standard error of the mean (S.E.M.).

[*] $p < 0.05$ and

[**] $p < 0.01$ vs control microbiota + vehicle group.

[#] $p < 0.05$ and

[##] $p < 0.01$ vs control microbiota + omeprazole group.

In terms of β-diversity metrics, PCoA analysis and PERMANOVA analysis showed that there was a significant difference in β-diversity of small intestinal microbiota between control mice and rebamipide-treated mice. Moreover, in the present study, in omeprazole-administered animals, there was significant difference in β-diversity between animals transplanted with control microbiota and animals transplanted with rebamipide-modulated microbiota. From the viewpoint of β-diversity, these results suggest that rebamipide has the potential to modulate small intestinal microbiota resistant to NSAID-induced small intestinal microbiota with or without PPI.

*Firmicutes*/*Bacteroidetes* (F/B) or B/F ratio is a relevant marker of dysbiosis. Accumulating evidence suggest the significance of F/B or B/F ratio as a biomarker of dysbiosis in the pathophysiology of various diseases such as obesity [31], diabetes [32], and multiple sclerosis [33]. In gastrointestinal mucosal damage, it is reported that Huai hua san, a traditional Chinese herbal formula, alleviated dextran sulphate sodium (DSS)-induced large intestinal mucosal damage with modulation of F/B ratio [44]. Our present study showed that omeprazole markedly decreased B/F ratio and transplantation of rebamipide-modulated microbiota recovered it. In the pathophysiology of NSAID-induced small intestinal damage, it is possible that F/B or B/F is useful marker for the assessment of NSAID-induced small intestinal damage-associated dysbiosis.

In the present study, we performed a comprehensive analysis of the alteration of the microbial composition of the small intestine by rebamipide and omeprazole at the species level using 16S rRNA gene sequencing analysis. The issue to be clarified is the determination of the responsible alterations of microbial composition, which modifies the pathophysiology of NSAID-induced small intestinal damage in the administration of rebamipide and PPI. In this study, in control microbiota-transplanted mice, omeprazole increased the proportion of *L. taiwanensis*, while transplantation of rebamipide-modulated microbiota significantly reduced the proportion of *L. taiwanensis*. Recently it is reported that *L. taiwanensis* produce bacteriocins, an antimicrobial peptide [45]. In the present study, increase in concentration of bacteriocins produced by excessive amount of *L. taiwanensis* may modulate small intestinal microbiota. However, there is little information about *L. taiwanensis*-derived bacteriocin in physiological and pathophysiological condition the small intestine, and impact of *L. taiwanensis*-derived bacteriocin on the component of microbiota remains unknown. In literature, accumulating studies indicate that *Lactobacillus*-derived bacteriocin exerts inhibitory effect against dysbiosis [46–48]. Considering these reports, it is possible that the result of reduction of proportion of *L. taiwanensis* is just the accompanying phenomenon and *L. taiwanensis*-derived bacteriocin is irrelevant to the effect of rebamipide on PPI-induced exacerbation of NSAID-induced small intestinal damage.

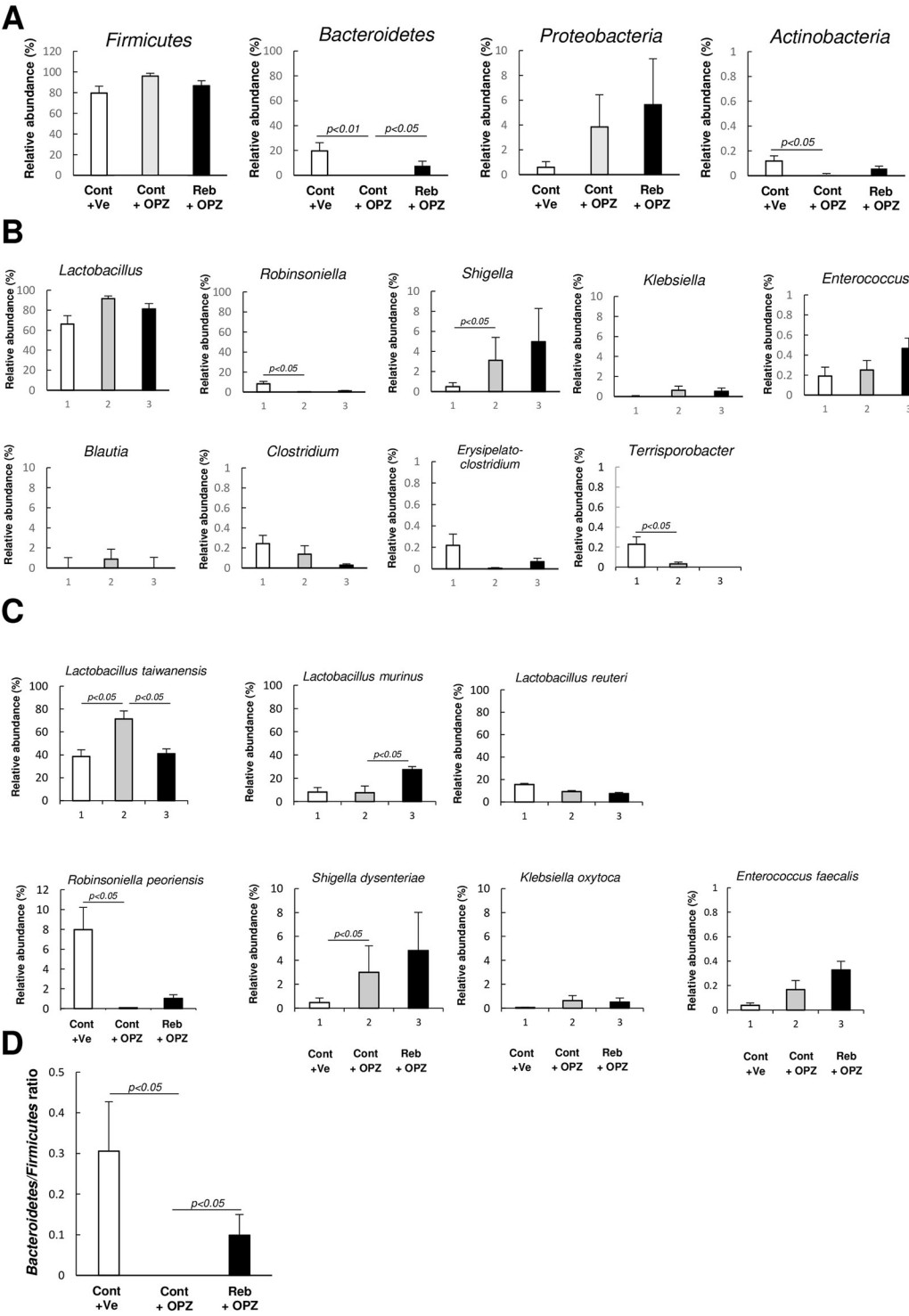

**Fig 6.** The major bacterial composition of small intestine in microbiota-transplanted mice administered with omeprazole or vehicle at (A) phylum level (B) genus level (C) species level. (D) *Bacteroidetes* to *Firmicutes* ratio. Cont + Ve: vehicle-administered group transplanted with control microbiota, Cont + OPZ: omeprazole-administered group transplanted with control microbiota, Reb + OPZ: omeprazole-administered group transplanted with rebamipide-modulatd microbiota. Each column represents the mean ± standard error of the mean (S.E.M.). $N$ = 6–7.

In our present study, rebamipide increased the proportion of *L. murinus*, even in the presence of the administration of omeprazole. The accumulated evidence suggested that *L. murinus* protected rats from necrotizing enterocolitis [49] and DSS-induced colitis in mice [50], and that *L. murinus* had anti-inflammatory properties in a human intestinal epithelial cell lines (Caco-2 cells) and it also improved gut barrier function in a mice model [51]. Our previous study revealed that supplementation of *L. murinus* ameliorated indomethacin-induced small intestinal damage [14]. These results indicate that it is possible that rebamipide increased the percentage of *L. murinus* in the small intestinal microbiota, which resulted in a suppressive effect on indomethacin-induced small intestinal damage and its exacerbation by omeprazole. This issue warrants further investigation.

Other statistically significant alteration of microbial composition in the small intestine by omeprazole and rebamipide was observed in the present study, such as the reduction of *Bacteroidetes* and *Actinobacteria* by omeprazole and the attenuation of the alteration by rebamipide; however, these were minor populations of microbial compositions of small intestine, and the pathophysiological impact on NSAID-induced small intestinal damage remains unknown.

There are several limitations to the present study. First, we cannot precisely evaluate the impact of each of alteration of the microbial component; even alteration of minor microbial component may have great impact on the pathophysiology of NSAID-induced small intestinal damage. To evaluate them, usage of gnotobiotic mice is necessary.

Second, we cannot exclude the possibility of the significance of residual rebamipide in the transplanted ileal contents. To minimize the possibility, we put 24 hours of washout period before transplantation and 4 days period before indomethacin challenge in the experiment protocol. Even though, it is possible that there may be small amount of residual rebamipide in the ileal content transferred with donor microbiota to the recipient mice, and the residual rebamipide may exerts direct pharmacological effect on the intestinal mucosa of the recipient mice. To clear the possibility, the analysis of the detection of residual rebamipide in the ileal contents obtained from donor mice may be useful, however, it is impossible for us because of technical problem; there is no commercial assay kit and hand-made laboratory method to determine the amount of rebamipide in the ileal contents is not established. Third, accumulating evidence is establishing the dysbiosis in the large intestine, while it remains unknown about the dysbiosis in the pathophysiology of NSAID-induced small intestinal damage. For example, B/F ratio is originally determined in the large intestinal microbiota and is well studied in the field of the pathophysiology of obesity [31]. However, it remains unclear about the significance of B/F or F/B ratio in the pathophysiology of NSAID-induced small intestinal damage.

In conclusion, rebamipide has potential to modulate small intestinal microbiota resistant to indomethacin-induced small intestinal damage. The suppressive effect extends to the suppression of exacerbation of indomethacin-induced small intestinal damage by proton pump inhibitors.

## Supporting information

**S1 Table. The major bacterial composition of small intestine in mice given rebamipide or vehicle at phylum level.**
(DOCX)

**S2 Table. The major bacterial composition of small intestine in mice given rebamipide or vehicle at genus level.**
(DOCX)

**S3 Table. The major bacterial composition of small intestine in mice given rebamipide or vehicle at species level.**
(DOCX)

**S4 Table. The major bacterial composition of small intestine in microbiota-transplanted mice administered with omeprazole at phylum level.**
(DOCX)

**S5 Table. The major bacterial composition of small intestine in microbiota-transplanted mice administered with omeprazole at genus level.**
(DOCX)

**S6 Table. The major bacterial composition of small intestine in microbiota-transplanted mice administered with omeprazole at species level.**
(DOCX)

## Acknowledgments

We thank Emi Suzuki-Yoshioka for her technical assistance.

## Author Contributions

**Conceptualization:** Tetsuya Tanigawa.

**Data curation:** Tetsuya Tanigawa, Sunao Shimada.

**Formal analysis:** Tetsuya Tanigawa, Sunao Shimada, Wataru Suda, Masahira Hattori.

**Investigation:** Tetsuya Tanigawa, Akira Higashimori, Sunao Shimada, Hiroyuki Kitamura, Takuya Kuzumoto, Wataru Suda, Masahira Hattori.

**Methodology:** Tetsuya Tanigawa, Akira Higashimori, Sunao Shimada, Hiroyuki Kitamura, Takuya Kuzumoto, Wataru Suda, Masahira Hattori.

**Project administration:** Tetsuya Tanigawa.

**Supervision:** Toshio Watanabe, Masahira Hattori, Yasuhiro Fujiwara.

**Validation:** Toshio Watanabe, Yuji Nadatani, Koji Otani, Shusei Fukunaga, Shuhei Hosomi, Fumio Tanaka, Noriko Kamata, Yasuaki Nagami, Koichi Taira, Masatsugu Shiba.

**Writing – original draft:** Tetsuya Tanigawa.

**Writing – review & editing:** Tetsuya Tanigawa.

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
