## [Decision Letter · Decision Letter 0]

10 Sep 2020

PONE-D-20-23408

Rebamipide Ameliorates Indomethacin-Induced Small Intestinal Damage and Proton Pump Inhibitor-Induced Exacerbation of this Damage by Modulation of Small Intestinal Microbiota

PLOS ONE

Dear Dr. Tanigawa,

Thank you for submitting your manuscript to PLOS ONE. After careful consideration, we feel that it has merit but does not fully meet PLOS ONE’s publication criteria as it currently stands. Therefore, we invite you to submit a revised version of the manuscript that addresses the points raised during the review process.

As you will see below, the reviewers highlighted several deficiencies in the methods that need to be addressed. Furthermore, the manuscript requires attention to address the grammatical and spelling concerns highlighted by the reviewers.

We look forward to receiving your revised manuscript.

Kind regards,

Jan S Suchodolski, DVM, PhD

Academic Editor

PLOS ONE

Journal Requirements:

2. Please ensure that all statements are justified and/or referenced appropriately. For example, in the Introduction section, there are numerous statements which appear to be currently unsubstantiated and require a reference. For example “The emergence of video capsule endoscopy and balloon-assisted enteroscopy have enabled the precise diagnosis of diseases involved in small intestinal damage, and the clinical features of nonsteroidal anti-inflammatory drug (NSAID)-induced small intestinal damage have been clarified” and “NSAIDs frequently induce small intestinal damage, which cause overt and occult bleeding, perforation, and stenosis of the small intestine".

Reviewers' comments:

Reviewer's Responses to Questions

**Comments to the Author**

1. Is the manuscript technically sound, and do the data support the conclusions?

Reviewer #1: Yes

Reviewer #2: Partly

2. Has the statistical analysis been performed appropriately and rigorously? 

Reviewer #1: Yes

Reviewer #2: Yes

3. Have the authors made all data underlying the findings in their manuscript fully available?

Reviewer #1: No

Reviewer #2: No

4. Is the manuscript presented in an intelligible fashion and written in standard English?

Reviewer #1: Yes

Reviewer #2: Yes

5. Review Comments to the Author

Reviewer #1: Interesting and well written manuscript. The study elucidates one mechanism through which rebapimide has a protective effect against indomethacin-induced small intestinal damage, even in the presence of omeprazole. One limitation of the study is the method of DNA extraction, which included a number of filtering and washing steps, followed by enzymatic lysis. While individually each one of those steps may have a minor impact, combined they introduce a significant bias which may have impacted the detection of microbial diversity. However, given that the protocol is thoroughly described and was equally applied to all samples, the results are still valid and applicable.

Specific comments:

Table numbers: There is no table 3, please renumber the tables accordingly.

Tables 4 and 8: It would be interesting to calculate the Firmicutes:Bacteroidetes ratio, an indicator of dysbiosis commonly used in the study of obesity and neurologic diseases. Based on your means in table 8 I believe it will be significantly different between treatments, and it is an additional indicator that will strengthen your discussion.

At the beginning of page 22 (paragraph before table 10) you mention that omeprazole decreased L. murinus. Based on table 10 that is not correct, as the change is of only 0.5% and non-significant. Please correct. Accordingly, correct that L. murinus was non ameliorated but significantly increased by rebamipide.

Page 23:

On the sentence after reference 29, please italicize L. taiwanensis.

It would be interesting to discuss the bacteriocin production further – that is a potential explanation for the reduced diversity in the omeprazole-treated mice.

Reviewer #2: This study aims to determine the ability of rebamipide conditioned microbiota to protect against NSAID-induced GI injury and omeprazole exacerbated NSAID-induced GI injury. The study design is simple (although many important details are not included) and involves ileal microbiota transplant of rebamipide treated mice to recipients that are then treated with indomethacin and indomethacin plus omeprazole. The limited readouts include histology, macroscopic injury scores, and basic microbiota analysis. There are numerous grammatical errors but overall it is well-written.

There is minimal new information in this simple study and the lack of details in M and M, lack of better control groups, and minimal readouts prevent robust evaluation.

where are the sequecing data available?

Please include more information in M and M section regarding animal studies. What were mice fed? How were they housed (all groups separately from shipping through study? all treatment groups in individual cages, etc...)?

Use of isoflurane is uncecessarily repeated.

What is deep euthanasia?

How was indomethacin administered? "Orally" is not clear. Gavage?

How were sections of SI that "that exhibited typical

mucosal damage" selected? Was the person who selected these blinded to treatment group?

Was the ileal contents from the donor mice stored or given fresh. If stored, how was it stored?

It is stated that the ileal contents were dissolved. This seems incorrect as I do not believe lumenal contents are soluble.

Suggest: ...intestinal damage was evaluated after 24 h.

In the second series of experiments it is stated that "the mice were treated with antibiotics for 7 d" same antibiotics as first study? If so this should be clearly stated.

Figure 1 is helpful. Typo in figure 1 (anpicillin)

How many mice in each group?

How was omep administered?

what do you mean "The content of the small intestinal lumen was collected immediately after sampling

from the mice."?

typo - In brief, prior to DNA extraction, each sample suspended in 15 mL PBS buffer

(Thermo Fisher Scientific K.K., Tokyo, Japan) was filtered ...

In this sentence...twice using a glass or plastic bar with PBS

buffer..How much PBS? what is a bar? why switch between glass and plastic?

The complete computational analysis of 16S data is lacking. For example, how was PCoA generated?

How was mm^2 area of evans blue staining generated (how was area of evans blue staining quantified?

Table 2 could be made much more visually appealing and easier to follow.

Numerous grammatical errors throughout, far too many to correct. One example, "There is no difference in OTU number in"... present tense here. past tense elsewhere in results.

There are many limitations in this manuscript that should be discussed.

How can you be sure that the positive effects seen are not due to residual rebemipide in the ileal contents?

Inclusion of a control group not being treated with indomethacin would greatly aid in interpretation of these data. For example, perhaps rebamipide had no effect on the microbiota but instead reduced GI injury via another mechanism and the modest differences in the microbiota were related to less inflammation and not rebamipides influences on the microbiota.

Please discuss beta diversity findings in more detail.

The comparisons and where/which significant differiences exist is not clear to me in tables 4-6 and 8-10

6. PLOS authors have the option to publish the peer review history of their article (what does this mean?). If published, this will include your full peer review and any attached files.

Reviewer #1: No

Reviewer #2: No

---

## [Author Response · Author response to Decision Letter 0]

16 Nov 2020

PONE-D-20-23408

PLOS ONE Decision: Major Revision

“Rebamipide Ameliorates Indomethacin-Induced Small Intestinal Damage and Proton Pump Inhibitor-Induced Exacerbation of this Damage by Modulation of Small Intestinal Microbiota”

Response to Reviewers and Editors

Editor’s comment

2. Please ensure that all statements are justified and/or referenced appropriately. For example, in the Introduction section, there are numerous statements which appear to be currently unsubstantiated and require a reference. For example “The emergence of video capsule endoscopy and balloon-assisted enteroscopy have enabled the precise diagnosis of diseases involved in small intestinal damage, and the clinical features of nonsteroidal anti-inflammatory drug (NSAID)-induced small intestinal damage have been clarified” and “NSAIDs frequently induce small intestinal damage, which cause overt and occult bleeding, perforation, and stenosis of the small intestine".

(Authors’ response)

We feel deeply sorry about the lack of appropriate references supporting the statement. We added them in the revised manuscript. 

3. Have the authors made all data underlying the findings in their manuscript fully available?

Reviewer #1: No

Reviewer #2: No

(Authors’ response)

We put the raw dataset on the “figshare”. Now all dataset underlying the findings in our revised manuscript fully available in figshare (https://figshare.com, DOI: 10.6084/m9.figshare.13241267).

5. Review Comments to the Author

Reviewer #1: Interesting and well written manuscript. The study elucidates one mechanism through which rebapimide has a protective effect against indomethacin-induced small intestinal damage, even in the presence of omeprazole. One limitation of the study is the method of DNA extraction, which included a number of filtering and washing steps, followed by enzymatic lysis. While individually each one of those steps may have a minor impact, combined they introduce a significant bias which may have impacted the detection of microbial diversity. However, given that the protocol is thoroughly described and was equally applied to all samples, the results are still valid and applicable.

Specific comments:

(1) Table numbers: There is no table 3, please renumber the tables accordingly.

(Authors’ response)

We feel terribly sorry about our careless mistake. We renumbered the tables accordingly. 

(2) Tables 4 and 8: It would be interesting to calculate the Firmicutes:Bacteroidetes ratio, an indicator of dysbiosis commonly used in the study of obesity and neurologic diseases. Based on your means in table 8 I believe it will be significantly different between treatments, and it is an additional indicator that will strengthen your discussion.

(Authors’ response)

We really appreciate the reviewer’s suggestion. Because of the mathematical problem (“division by zero” problem, because in some mice the percentage of Bacteroidetes was “zero”), alternatively we calculated Bacteroidetes/Firmicutes ratio. We added the data and discussion about the data with relevant references. 

(3) At the beginning of page 22 (paragraph before table 10) you mention that omeprazole decreased L. murinus. Based on table 10 that is not correct, as the change is of only 0.5% and non-significant. Please correct. Accordingly, correct that L. murinus was non ameliorated but significantly increased by rebamipide.

(Authors’ response)

We appreciate the reviewer’s advice and we agree to it. It is our careless misunderstanding. We revised the sentences accordingly. 

(4) Page 23: On the sentence after reference 29, please italicize L. taiwanensis.

(Authors’ response)

We appreciate the reviewer’s advice. We italicized the word, “L. taiwanensis.” We checked again the manuscript and italicized all of the bacterial names. 

(5) It would be interesting to discuss the bacteriocin production further – that is a potential explanation for the reduced diversity in the omeprazole-treated mice.

(Authors’ response)

We appreciate the reviewer’s advice. In accordance with the reviewer’s advice, we discuss the bacteriocin production more profoundly. 

Reviewer #2: This study aims to determine the ability of rebamipide conditioned microbiota to protect against NSAID-induced GI injury and omeprazole exacerbated NSAID-induced GI injury. The study design is simple (although many important details are not included) and involves ileal microbiota transplant of rebamipide treated mice to recipients that are then treated with indomethacin and indomethacin plus omeprazole. The limited readouts include histology, macroscopic injury scores, and basic microbiota analysis. There are numerous grammatical errors but overall it is well-written.

(1) There is minimal new information in this simple study and the lack of details in M and M, lack of better control groups, and minimal readouts prevent robust evaluation.

(Authors’ response)

We deeply appreciate the editors’ and the reviewers’ patience to check my grammatical errors, typos, the lack of details in Materials and Methods, the lack of necessary references, and other careless mistakes. We would like to apologize all of them, and we corrected them. 

(2) Where are the sequencing data available?

(Authors’ response)

We put the raw dataset on the “figshare”. Now all dataset underlying the findings in our revised manuscript fully available in figshare (https://figshare.com, DOI: 10.6084/m9.figshare.13241267).

(3) Please include more information in M and M section regarding animal studies. What were mice fed? How were they housed (all groups separately from shipping through study? all treatment groups in individual cages, etc...)?

(Authors’ response)

We are terribly sorry for the reviewer’s confusion about the animal experiments of the present study. We added more information about the animal experiment in Materials and Methods section. 

(4) Use of isoflurane is uncecessarily repeated.

(Authors’ response)

We appreciate the reviewer’s suggestion. We revised the portion of the manuscript. 

(5) What is deep euthanasia?

(Authors’ response)

We appreciate the reviewer’s suggestion. We feel sorry for simple careless mistake which confuse the reviewers. That should be “deep sedation”. We revised the portion of the manuscript. 

(6) How was indomethacin administered? "Orally" is not clear. Gavage?

(Authors’ response)

We administered indomethacin to mice by gavage tube. We revised the portion. 

(7) How were sections of SI that "that exhibited typical mucosal damage" selected? Was the person who selected these blinded to treatment group?

(Authors’ response)

We appreciate the reviewer’s suggestion. We added the additional information about the Materials and Methods in the revised manuscript. 

(8) Were the ileal contents from the donor mice stored or given fresh? If stored, how was it stored?

(Authors’ response)

The ileal contents from the donor mice were given fresh to the recipient mice. 

(9) It is stated that the ileal contents were dissolved. This seems incorrect as I do not believe luminal contents are soluble.

(Authors’ response)

We appreciate the reviewer’s suggestion. It is just our careless mistake. We revised it (dissolved>>>suspended). 

(10) Suggest: ...intestinal damage was evaluated after 24 h.

(Authors’ response)

We feel terribly sorry for the grammatical error. We revised it. 

(11) In the second series of experiments it is stated that "the mice were treated with antibiotics for 7 d" same antibiotics as first study? If so this should be clearly stated.

(Authors’ response)

We used the same antibiotics as first study. We revised the manuscript to describe it clearly in the Materials and Methods section. 

(12) Figure 1 is helpful. Typo in figure 1 (anpicillin)

(Authors’ response)

We really appreciate the reviewer’s point out and we are terribly sorry for the typo. We revised it. 

(13) How many mice in each group?

(Authors’ response)

For molecular biological analysis, we used 6-8 animals in each group and for assessment of macroscopic and microscopic experiments we used 5-7 animals. It is described in the Materials and Methods and Figure Legends. 

(14) How was omep administered?

(Authors’ response)

We are terribly sorry for our immature description. Omeprazole was orally administered with gavage tube. We revised the Materials and Methods section. 

(15) What do you mean "The content of the small intestinal lumen was collected immediately after sampling from the mice."?

(Authors’ response)

We are terribly sorry for our immature description. We revised it as “The content of the small intestinal lumen was collected immediately after removing the intestine from the mice”.

(16) typo - In brief, prior to DNA extraction, each sample suspended in 15 mL PBS buffer (Thermo Fisher Scientific K.K., Tokyo, Japan) was filtered ...

(Authors’ response)

We appreciate the reviewer’s indication. We revised the manuscript. 

(17) In this sentence...twice using a glass or plastic bar with PBS buffer..How much PBS? what is a bar? why switch between glass and plastic?

(Authors’ response)

We feel sorry for the reviewer’s confusion. We comfirmed the truth to the technical assistant. Actually we used a plastic bar. We revised it. 

(18) The complete computational analysis of 16S data is lacking. For example, how was PCoA generated?

(Authors’ response)

We really appreciate the reviewer’s suggestion. We revised the manuscript to make it clear. 

(19) How was mm^2 area of evans blue staining generated (how was area of Evans blue staining quantified?

(Authors’ response)

The shape of macroscopic mucosal break was round or elliptical shape, therefore we measured the major axis and minor axis with a digital precision caliper, summed for the small intestine, and used the value as the lesion index. In accordance with the reviewer’s query, we revised the portion to describe the detail. 

(20) Table 2 could be made much more visually appealing and easier to follow.

(Authors’ response)

We really appreciate the reviewer's advice. We considered the data again and we omitted the unnecessary comparison in the Table. 

(21) Numerous grammatical errors throughout, far too many to correct. One example, "There is no difference in OTU number in"... present tense here. past tense elsewhere in results.

(Authors’ response)

We feel very sorry for your check of our manuscript with numerous grammatical errors. We checked the manuscript again extensively and we revised it. 

(22) There are many limitations in this manuscript that should be discussed.

(Authors’ response)

We agree to the reviewer’s suggestion and all of the reviewer’s suggestion is very helpful to consider the limitations in the present study. We discuss the limitation to the present study in Discussion section. 

(23) How can you be sure that the positive effects seen are not due to residual rebemipide in the ileal contents?

(Authors’ response)

We understand the intention of the reviewer’s comment. To avoid the effect of residual rebamipide in the ileal contents, we put 24 hours of washout period before transplantation and 4 days period before indomethacin challenge in the experiment protocol. Even though, it is possible that there may be small amount of residual rebamipide in the ileal content transferred with donor microbiota to the recipient mice and the residual rebamipide may exerts direct pharmacological effect on the small intestine of the recipient mice independent of modulation of small intestinal microbiota. To clear the possibility, the detection assay of residual rebamipide in the ileal content obtained from donor mice may be useful, however, it is impossible for us because of technical problem; there is no commercial assay kit and hand-made laboratory method to determine the amount of rebamipide in the ileal contents is not established. We describe the issue as the limitation to the present study in the Discussion session. 

(24) Inclusion of a control group not being treated with indomethacin would greatly aid in interpretation of these data. For example, perhaps rebamipide had no effect on the microbiota but instead reduced GI injury via another mechanism and the modest differences in the microbiota were related to less inflammation and not rebamipides influences on the microbiota.

(Authors’ response)

In the analysis of the microbial composition, all animals were not treated with indomethacin to avoid the influence of indomethacin-induced mucosal damage. 

(25) Please discuss beta diversity findings in more detail.

(Authors’ response)

We discuss beta diversity findings in more detail. 

(26) The comparisons and where/which significant differences exist is not clear to me in tables 4-6 and 8-10. 

(Authors’ response)

To make the comparisons and significant differences clear, we made the figures for the main results. Tables are also submitted as the supplementary data.

---

## [Editor Report · Decision Letter 1]

12 Jan 2021

Rebamipide Ameliorates Indomethacin-Induced Small Intestinal Damage and Proton Pump Inhibitor-Induced Exacerbation of this Damage by Modulation of Small Intestinal Microbiota

PONE-D-20-23408R1

Dear Dr. Tanigawa,

We’re pleased to inform you that your manuscript has been judged scientifically suitable for publication and will be formally accepted for publication once it meets all outstanding technical requirements.

Kind regards,

Jan S Suchodolski, DVM, PhD

Academic Editor

PLOS ONE
---

## [Editor Report · Acceptance letter]

20 Jan 2021

PONE-D-20-23408R1 

Rebamipide Ameliorates Indomethacin-Induced Small Intestinal Damage and Proton Pump Inhibitor-Induced Exacerbation of this Damage by Modulation of Small Intestinal Microbiota 

Dear Dr. Tanigawa:

I'm pleased to inform you that your manuscript has been deemed suitable for publication in PLOS ONE. Congratulations! Your manuscript is now with our production department. 

Kind regards, 

on behalf of

Dr. Jan S Suchodolski 

Academic Editor

PLOS ONE